# ATTACK-RESISTANT FEDERATED LEARNING WITH RESIDUAL-BASED REWEIGHTING

## ABSTRACT

Federated learning has a variety of applications in multiple domains by utilizing private training data stored on different devices. However, the aggregation process in federated learning is highly vulnerable to adversarial attacks so that the global model may behave abnormally under attacks. To tackle this challenge, we present a novel aggregation algorithm with residual-based reweighting to defend federated learning. Our aggregation algorithm combines repeated median regression with the reweighting scheme in iteratively reweighted least squares. Our experiments show that our aggregation algorithm outperforms other alternative algorithms in the presence of label-flipping, backdoor, and Gaussian noise attacks. We also provide theoretical guarantees for our aggregation algorithm.

## 1 INTRODUCTION

Federated learning is a machine learning methodology for training a global model with decentralized data stored on multiple or even millions of devices (McMahan et al., 2017). In federated learning, private data is stored locally in isolated devices and will not be revealed to other parties during training. Federated learning can enable numerous real-world machine learning applications by utilizing massive training data that are privacy-sensitive and scattered on different devices (Bonawitz et al., 2017). For instance, multiple hospitals can collaborate to train a global model for classifying diseases using X-ray images without compromising patient privacy. Note that these hospitals may possess X-ray images in different quantities and varieties, resulting in the non-IID (independent and identically distributed) data distribution. Federated learning is different from distributed learning in the sense that the training data is often non-IID and we have no control over data distribution in federated learning.

The default federated learning aggregation algorithm *FedAvg* (McMahan et al., 2017) that takes the average of locally updated models is vulnerable to various attacks. We find that federated learning suffers from label-flipping, backdoor, and Gaussian noise attacks in our experiments. When a local model is poisoned, the aggregated global model can also be poisoned and fail to behave correctly. A label-flipping attack (Biggio et al., 2012) happens where an attacker assigns incorrect labels to some data. For example, an attacker can train a local model with cat images mislabelled as dogs and then share the poisoned local model for aggregation.

Mitigating attacks in federated learning or distributed learning has been explored in recent research (Chen et al., 2017; Yin et al., 2018; Fung et al., 2018; Blanchard et al., 2017). Although the median or trimmed mean aggregation algorithms (Yin et al., 2018) may seem plausible in distributed learning, their performance degrades in federated learning when data is non-IID. FoolsGold (Fung et al., 2018) is a defense algorithm that identifies participants with similar models as attackers but this strategy may not work when some harmless participants have similar local data. To make federated learning more attack-resistant, we develop an aggregation algorithm that is robust against label-flipping, backdoor, and Gaussian noise attacks in a general non-IID setting. We derive our aggregation algorithm by adopting the repeated median estimator (Siegel, 1982) and the reweighting scheme in iteratively reweighted least squares (IRLS) (Holland and Welsch, 1977; Rand R, 1997). We estimate the confidence of each parameter in the local models and then the weight of each local model can be computed by heuristically accumulating all the parameter confidence in each local model. Our algorithm is straightforward to implement. Furthermore, we provide theoretical guarantees for our aggregation algorithm.

We compare our proposed algorithm to several baselines by conducting experiments on four datasets, the MNIST dataset (LeCun et al., 1998), CIFAR-10 dataset (Krizhevsky et al., 2009), Amazon Reviews dataset (Ruining and Julian, 2016) and the Lending Club loan dataset (Kan, 2019). Our proposed aggregation significantly mitigates the impact of attacked models in non-IID federated learning and outperforms other baselines in our evaluation.

## 2 RELATED WORK

**Adversarial attacks on federated learning.** Several attacks have been studied against federated learning (Wang et al., 2018; Biggio et al., 2012; Fung et al., 2018; Hayes et al., 2019; Hitaj et al., 2017; Melis et al., 2019). The label-flipping attack (Biggio et al., 2012) is shown to have great harm to a federated system even with a very small number of attackers (Fung et al., 2018). In this attack, the attacker flips the labels of training data in one class to another class and trains the model accordingly. Bagdasaryan et al. (2018) propose a backdoor attack so that the global model behaves incorrectly on adversarial targeted input. In our work, we mainly focus on defending against label-flipping attack, backdoor, and Gaussian noise attacks. Note that an attacker can perform any type of attacks, such as modifying any model values and training the local model on poisoned data for arbitrary epochs.

**Robust distributed learning.** Statistical methods have been studied and applied in robust distributed learning where data is IID (Feng et al., 2014; Blanchard et al., 2017; Chen et al., 2017; Yin et al., 2018; Alistarh et al., 2018). The median method and the trimmed mean method (Yin et al., 2018) are effective approaches in robust distributed learning, but may not be attack-resistant in federated learning where data distribution is non-IID. To tackle the challenge in robust federated learning, we propose a reweighted aggregation algorithm that dynamically assigns weights to the local model based on the residual to a regression line estimated by the repeated median estimator (Siegel, 1982).

**Defending federated learning.** Recently, some researchers have proposed some defense strategies for robust federated learning (Fung et al., 2018; Blanchard et al., 2017). FoolsGold (Fung et al., 2018) is a defense mechanism against Sybil attacks by adjusting the learning rates of local models based on contribution similarity. The algorithm identifies grouped actions as Sybil attacks and promotes the diversity of local model update. However, FoolsGold may identify harmless participants as attackers when these participants have similar local data. Gu et al. (2018) proposed a model, CalTrain, that represents data with fingerprints to identify poisoned data and models. Konstantinov and Lampert (2019) propose to maintain a small reference dataset to justify the quality and accountability of models. While this method is effective, it requires a lot of time to evaluate each model in every single round. Our algorithm does not need an additional reference dataset before or after each aggregation process.

Some researchers proposed to improve the privacy preservation of federated learning (Bonawitz et al., 2017; Geyer et al., 2017; Truex et al., 2018; Thakkar et al., 2019). Bonawitz et al. (2017) propose a privacy-preserving protocol for model aggregation in federated learning. Geyer et al. (2017) introduce differential privacy into federated learning. Instead of enhancing privacy preservation, we focus on the robustness of federated learning so that the global model should behave correctly even when there is a large portion of malicious participants.

## 3 OUR ALGORITHM

In federated learning, there are multiple rounds of communication between participants and a central server for learning a global model. In each round, the global model is shared among the $K$ participants and a local model on each device is trained on its local private data with the shared global model as initialization. Then all the $K$ local models are sent to the central server to update the global model with an aggregation algorithm. The original aggregation scheme uses a simple averaging algorithm to aggregate all the local models (McMahan et al., 2017). Suppose the participant $k$ has a local model $M^{(k)}$, and we can update the global model $M_{global}$ by taking the (weighted) average of all the $K$ local models.

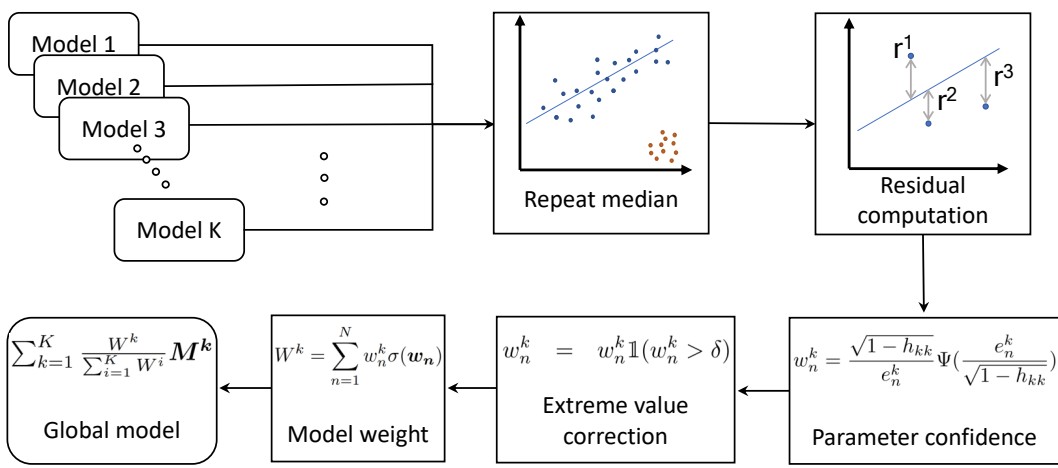

Figure 1: The overview of our aggregation algorithm for attack-resistant federated learning.

## 3.1 Aggregation Algorithm

Median is a robust estimator widely used in statistics. However, when the data distribution is non-IID, median neglects a significant amount of information by merely taking a single median value. Hence, in our aggregation algorithm, the global model is designed to be a reweighted average of all the local models where the model weights are estimated robustly.

Algorithm 1 summarizes our aggregation algorithm, and a detailed step-by-step description is provided below. We perform a weighted average of all the local models at the model level by assigning a weight to each local model. The weight of each local model is computed by accumulating the parameter confidence in the local model. The parameter confidence is computed based on the residual to a regression line estimated by the repeated median estimator (Siegel, 1982; Rand R, 1997). Inspired by the reweighting scheme in IRLS (Rand R, 1997), we reweight each parameter by its vertical distance (residual) to a robust regression line. For the robust regression line estimation, we use the repeated median estimator (Siegel, 1982) since it has a high breakdown point of 50%.

Let $y_n^{(k)}$ be the $n$-th parameter of the $k$-th local model. We use $\boldsymbol{y_n}$ to indicate the list of $n$-th parameters in all the local models. Let $\boldsymbol{x_n}$ be the indices of $\boldsymbol{y_n}$ sorted in an ascending order. Then $(\boldsymbol{x_n}, \boldsymbol{y_n})$ is a point set in 2D with increasing values in the $y$ direction.

**Repeated median.** We use the repeated median estimator (Siegel, 1982) to estimate a linear regression line $y = \beta_{n0} + \beta_{n1}x$. The slope $\beta_{n1}$ and intercept $\beta_{n0}$ are estimated as follow,

$$\beta_{n1} = \underset{i}{\text{median}}\,\underset{i \neq j}{\text{median}}\, \frac{y_n^{(j)} - y_n^{(i)}}{x_n^{(j)} - x_n^{(i)}}, \quad i,j \in \{1, 2, ..., K\} \tag{1}$$

$$\beta_{n0} = \underset{i}{\text{median}}\,\underset{i \neq j}{\text{median}}\, \frac{x_n^{(j)} y_n^{(i)} - x_n^{(i)} y_n^{(j)}}{x_n^{(j)} - x_n^{(i)}}, \quad i,j \in \{1, 2, ..., K\} \tag{2}$$

**Residual computation.** We can calculate the residuals of the $n$-th parameters in all the local models:

$$\boldsymbol{r_n} = \boldsymbol{y_n} - \beta_{n0} - \beta_{n1}\boldsymbol{x_n}.$$

Since $\boldsymbol{r_n}$ can be very different in magnitude for different parameters, we can normalize $\boldsymbol{r_n}$ similar to the reweighting scheme in IRLS (Rand R, 1997):

$$\tau_n = \gamma \widetilde{|r_n|}(1 + \frac{5}{K-1}), \tag{3}$$

$$\widetilde{|r_n|} = \text{median}(|\boldsymbol{r_n}|). \tag{4}$$

where $\gamma$ is a constant. We set $\gamma = 1.48$ recommended by Wilcox et al. (Rand R, 1997). Then the normalized residuals become

$$e_n^{(k)} = \frac{r_n^{(k)}}{\tau_n}. \tag{5}$$

---

**Algorithm 1** Our aggregation algorithm

---

**Input:** Models $M^{(1)}, M^{(2)}, ..., M^{(K)}$, with parameters $y_n^{(1)}, y_n^{(2)}, ..., y_n^{(K)}$
**Output:** The global model $M_{global}$

1: **for** $n$-th parameter where $n = 1 \to N$, and let $\boldsymbol{y_n} = [y_n^{(1)}, y_n^{(2)}, ..., y_n^{(K)}]^T$ **do**
2:      $\boldsymbol{x_n} \leftarrow$ indices of $\boldsymbol{y_n}$ sorted in an ascending order          ▷ assign indices
3:      $\beta_{n0}, \beta_{n1} \leftarrow RepeatedMedian(\boldsymbol{x_n}, \boldsymbol{y_n})$          ▷ get robust line estimation
4:      $\boldsymbol{r_n} \leftarrow \boldsymbol{y_n} - \beta_{n0} - \beta_{n1}\boldsymbol{x_n}$          ▷ compute residual
5:      $\widetilde{|r_n|} \leftarrow Median|\boldsymbol{r_n}|$
6:      $\tau_n \leftarrow \gamma\widetilde{|r_n|}(1 + \frac{5}{K-1})$          ▷ normalize residuals
7:      $\boldsymbol{e_n} \leftarrow \frac{\boldsymbol{r_n}}{\tau_n}$
8:      $\boldsymbol{H_n} \leftarrow \boldsymbol{x_n}(\boldsymbol{x_n^T}\boldsymbol{x_n})^{-1}\boldsymbol{x_n^T}$          ▷ compute Hat matrix
9:      $\boldsymbol{w_n} \leftarrow \frac{\sqrt{1-diag(\boldsymbol{H_n})}}{\boldsymbol{e_n}}\Psi(\frac{\boldsymbol{e_n}}{\sqrt{1-diag(\boldsymbol{H_n})}})$          ▷ compute parameter confidence
10:      $\boldsymbol{y_n}, \boldsymbol{w_n} \leftarrow$ CorrectExtremeValue$(\boldsymbol{y_n}, \boldsymbol{w_n})$
11:      $\boldsymbol{w_n} \leftarrow \boldsymbol{w_n}\sigma(\boldsymbol{w_n})$          ▷ reweight confidence by its standard deviation
12: **for** each $k = 1 \to K$ **do**
13:      $W^{(k)} \leftarrow \sum_{n=1}^N \boldsymbol{w_n^{(k)}}$          ▷ accumulate weights
14: $M_{global} \leftarrow \sum_{k=1}^K \frac{W^{(k)}}{\sum_{i=1}^K W^{(i)}}M^{(k)}$          ▷ the global model aggregation

---

**Parameter confidence.** After obtaining the normalized residuals, the parameter confidence can be determined accordingly (Rand R, 1997):

$$w_n^{(k)} = \frac{\sqrt{1 - h_{kk}}}{e_n^{(k)}}\Psi(\frac{e_n^{(k)}}{\sqrt{1-h_{kk}}}), \quad (6)$$

where $w_n^{(k)}$ is the confidence of the $n$-th parameter in $M^{(k)}$, $\Psi(x) = max\{-Z, min(Z, x)\}$ with $Z = \lambda\sqrt{2/K}$ and $\lambda$ is a hyperparameter. We use $\lambda = 2$ in our experiments. $\Psi$ here acts as a trusted interval and we can expand or shrink the interval by tuning $\lambda$. $h_{kk}$ is the k-th diagonal element of matrix in $H_n$:

$$\boldsymbol{H_n} = \boldsymbol{x_n}(\boldsymbol{x_n^T}\boldsymbol{x_n})^{-1}\boldsymbol{x_n^T}, \quad (7)$$

where $\boldsymbol{x_n} = [x_n^{(1)} \ x_n^{(2)} \ \dots \ x_n^{(K)}]^T$.

**Extreme value correction.** Extremely large values, even multiplied with a small weight in

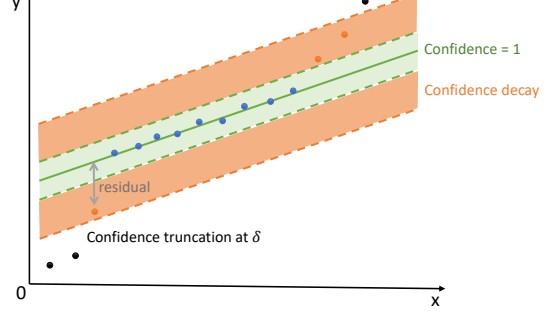

Figure 2: Parameter confidence assignment based on the residual which is the distance from a point to the regression line. In the green area, the parameter confidence is 1; in the orange area, the confidence decays from 1 to $\delta$; in other areas, the confidence is set to 0.

model aggregation, can damage the global model. We address this issue by involving a a threshold $\delta$. If a parameter has a confidence value lower than $\delta$, then it should be corrected as follows,

$$w_n^{(k)} = w_n^{(k)}\mathbb{1}(w_n^{(k)} > \delta), \quad (8)$$

$$y_n^{(k)} = y_n^{(k)}\mathbb{1}(w_n^{(k)} > \delta) + (\beta_{n0} + \beta_{n1}x_n^{(k)})\mathbb{1}(w_n^{(k)} \leq \delta). \quad (9)$$

The process of selecting proper $\delta$ is discussed in the appendix.

**Model weight.** To obtain the weight of each local model, we can simply aggregate the parameter confidence in the local model but this is not ideal. Imagine an attacker trains a model honestly, but then alters only 10% of the parameters to some extremely large values. This adversary model still receives about 90% of the parameter confidence. To address this problem, we measure the importance of a parameter by the standard deviation of $\boldsymbol{w_n}$. A confidence assignment with a large standard deviation indicates a great disagreement among this parameter in all models and should be more critical when being accumulated towards model weights:

$$W^{(k)} = \sum_{n=1}^N w_n^{(k)}\sigma(\boldsymbol{w_n}), \quad (10)$$

where $W^{(k)}$ is the weight for model $k$, $N$ is the number of parameters, $\boldsymbol{w_n} = [w_n^{(1)} w_n^{(2)} \ldots w_n^{(K)}]^T$.

**Global model.** Finally, we can obtain the updated global model by

$$M_{global} = \sum_{k=1}^{K} \frac{W^{(k)}}{\sum_{i=1}^{K} W^{(i)}} M^{(k)}. \tag{11}$$

## 3.2 THEORETICAL GUARANTEE

For simplicity, we consider the bound of the error rate for training a single-parameter model on $K$ devices, each storing $S$ IID samples of data. Suppose that there are $K$ devices and $U$ of them are corrupted. We denote the set of adversarial devices as $\mathcal{B}$ and the corruption ratio $\alpha = \frac{U}{K}$. We define the parameter of the local model $i$ as $\hat{y}^{(i)}$ and let $\mu$ be the optimal model that minimizes the population loss. Based on our algorithm, the residuals can be simplified as $(\hat{y}^{(i)} - \tilde{y})$, where $\tilde{y}$ is the median of $\{\hat{y}^{(i)}\}$ for $i = 1, 2, ..., K$. Let $\widetilde{|r|}$ be the median of absolute residuals, i.e., $\widetilde{|r|} = \text{median}(|\hat{y}^{(i)} - \tilde{y}|)$. Then, the normalized residual can be expressed as

$$e^{(i)} = \frac{\hat{y}^{(i)} - \tilde{y}}{\gamma \widetilde{|r|}(1 + \frac{5}{K-1})}, \tag{12}$$

and the parameter confidence is defined as

$$z^{(i)} = \begin{cases} 1, & \text{if } \left|e^{(i)}\right| \le \frac{\sqrt{2}\lambda}{\sqrt{K}} \\[2mm] \left|\dfrac{\sqrt{2}\lambda}{\sqrt{K}e^{(i)}}\right|, & \text{if } \dfrac{\sqrt{2}\lambda}{\sqrt{K}} < \left|e^{(i)}\right| \le \left|\dfrac{\sqrt{2}\lambda}{\sqrt{K}\delta}\right| \\[2mm] 0, & \text{if } \left|e^{(i)}\right| > \left|\dfrac{\sqrt{2}\lambda}{\sqrt{K}\delta}\right| \end{cases} \tag{13}$$

Then we will prove that the error of the global model $M_{global} := \frac{1}{\sum_{j=1}^{K} z^{(j)}} \sum_{i=1}^{K} z^{(i)} \hat{y}^{(i)}$ is bounded:

$$\left| \frac{1}{\sum_{j=1}^{K} z^{(j)}} \sum_{i=1}^{K} z^{(i)} \hat{y}^{(i)} - \mu \right| = \tilde{\mathcal{O}}\left( \frac{1}{\sqrt{S}} + \frac{1}{\sqrt{SK}} + \frac{1}{S} + \frac{1}{\sqrt{K}\delta} \right). \tag{14}$$

The details of the proof are presented in the appendix. The bound indicates that when the amount of data and the number of models are sufficiently large, we can achieve a small error rate $\tilde{\mathcal{O}}(\frac{1}{\sqrt{S}} + \frac{1}{\sqrt{SK}} + \frac{1}{S} + \frac{1}{\sqrt{K}\delta})$ with high probability.

## 4 EXPERIMENTS

We compare our approach with other aggregation algorithms, including FedAvg (McMahan et al., 2017), the coordinate-wise median method (Yin et al., 2018), the coordinate-wise trimmed mean method (Yin et al., 2018), FoolsGold (Fung et al., 2018), and a coordinate-wise repeated median approach we adopt from (Siegel, 1982). We perform experiments on the MNIST handwritten digit dataset (LeCun et al., 1998), the Amazon Reviews dataset (Ruining and Julian, 2016), the CIFAR-10 dataset (Krizhevsky et al., 2009), and the Lending Club loan dataset (Kan, 2019). We implement attack strategies and defense algorithms in PyTorch (Paszke et al., 2017).

We use a two-layer convolutional neural network (CNN) for our MNIST experiments and the network architecture is shown in the appendix. With this simple CNN model, our goal is to evaluate different aggregation algorithms for defending federated learning in the presence of attacks. On the CIFAR-10 dataset, we use ResNet-18 (He et al., 2015) for image classification. The text classification model *FastText* (Joulin et al., 2016) is adopted for evaluation on the Amazon Reviews dataset. It is a two-layer deep neural network where the first layer is an embedding layer, and the second layer is a fully connected layer. For the Lending Club loan dataset, we use a simple neural network with

three fully-connected layers to classify loan status. We use the last two models to demonstrate that our algorithm can be generalized to a natural language processing task and to a real-work financial problem. All the evaluation results are the average of running the same experiments 3 times.

## 4.1 DATASETS AND EXPERIMENTAL

**MNIST dataset.** The MNIST dataset contains 70,000 real-world handwritten images with digits from 0 to 9. We evaluate different methods by learning a global model on these training images distributed on multiple devices in a non-IID setting with adversarial attacks.

**CIFAR-10 dataset.** The CIFAR-10 dataset contains 60,000 natural images in ten object classes. The experimental setup is also non-IID on CIFAR-10.

**Amazon Reviews dataset.** The Amazon Reviews dataset (Ruining and Julian, 2016) contains product reviews and ratings collected from the Amazon website. Every review is paired with a sentiment rating from 1 to 5. We categorize comments with rating 1 or 2 as negative and comments with rating 4 or 5 as positive. We discard reviews with rating 3, and we only train a binary classifier. We only use the book reviews from the Kindle Store. 20% of the reviews are used for testing, while the rest is for training. We obtain a training set of 86,164 reviews and a test set of 13,260 reviews.

**Lending Club Loan dataset.** The Lending club dataset LOAN (Kan, 2019) contains financial information such as credit scores and the number of finance inquiries for loan status prediction ("Current", "Late", or "Fully Paid"). There are 1,808,534 data samples in 9 types of loan status. We divide them by US states to simulate the federated learning scenarios with non-IID data distribution where each state represents a participant.

## 4.2 RESULTS ON LABEL-FLIPPING ATTACKS

We evaluate the overall classification performance of different aggregation methods on three datasets under label-flipping attacks, the MNIST dataset (Biggio et al., 2012), the CIFAR-10 dataset (Krizhevsky et al., 2009), and Amazon Reviews dataset (Ruining and Julian, 2016). In label-flipping attacks, attackers flip the labels of training examples in the source class to a target class and train their models accordingly.

In the MNIST experiment, we simulate federated learning with 100 participants, within which 0 to 10 of them are attackers. Each participant contains images of two random digits. The attackers are chosen to be some participants with images of digit 1 and another random digit since they are flipping the label of 1 to 7. We run 200 synchronization rounds with $\delta$ set to 0.01. In each round of federated learning, each participant is supposed to train the local model for 5 epochs, but the attackers can train for arbitrary epochs. The results are shown in Figure 3 where attackers train the models with 5 more epochs to enhance the attacks. Our algorithm outperforms all other methods and is robust when the number of attackers increases. Median methods (Median and Repeated Median) have relatively low accuracy due to discarding most of the information in model aggregation. Our algorithm, on the other hand, takes the reweighted average of all local models and thus gathers more information in an unsupervised way. We also compare our algorithm with the state-of-the-art algorithm FoolsGold (Fung et al., 2018). Though their algorithm also maintains a low attack success rate, our algorithm is more stable and surpasses FoolsGold by 6% on average.

For the CIFAR-10 experiment, following Bagdasaryan et al. (2018), we adopt a Dirichlet distribution (Minka, 2000) with a hyperparameter 0.9 to generate non-IID data distribution for totally 10 participants. The experimental setup is the same for the CIFAR-10 and MNIST experiments under backdoor attacks in Section 4.3. The attackers flip the label of "cat" to "dog" since they are the most similar classes in CIFAR-10. The experiment results can be found in Table 1. In this experiment, baseline methods perform fairly well because the data distribution is not immensely non-i.i.d., where each user has all the labels but in different amounts. FoolsGold (Fung et al., 2018) fails because they assume that the gradients of honest models are very different because of the non-i.i.d. data, and the gradients of outliers are close because they share the same objective. However, it may not always be the case when the data distribution is not so extremely non-i.i.d, such as the CIFAR-10 case. FoolsGold may think these honest users are close and decide they are outliers. Our algorithm, on the other hand, can adapt to extremely non-i.i.d. cases, such as the MNIST experiments, where all

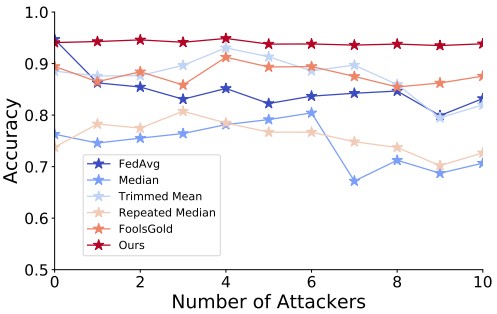
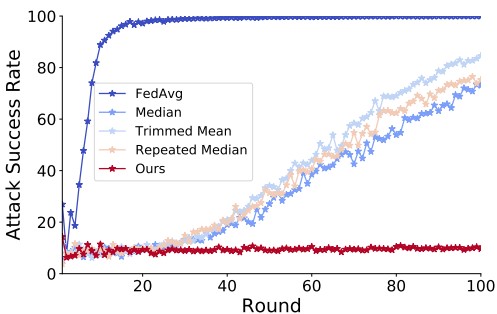

Figure 3: Results of label-flipping attacks on the MNIST dataset. The number of participants is 100, within which 0 to 10 of them are attackers.

Figure 4: Result of backdoor attack success rate on CIFAR-10 under different aggregation algorithms. Ours outperforms other baselines.

baseline methods fail, and is also suitable for common non-i.i.d. situations such as the CIFAR-10 and the Amazon Review experiments.

In the experiment on Amazon Reviews, there are 10 participants, where 0 to 4 participants are attackers who flip all their labels. Attackers also train for 5 more epochs. We run 10 synchronization rounds in the experiment. The result is summarized in Table 2. Our algorithm achieves comparable state-of-the-art results, and our performance does not degrade when there are less than 50% attackers.

| # of attackers | 0 | 1 | 2 | 3 | 4 | Average |
|---|---|---|---|---|---|---|
| FedAvg (McMahan et al., 2017) | 88.96% | 85.74% | 82.49% | 82.35% | 82.11% | 84.33% |
| Median (Yin et al., 2018) | 88.11% | 87.69% | 87.15% | 85.85% | 82.01% | 86.16% |
| Trimmed Mean (Yin et al., 2018) | 88.70% | 88.52% | **87.44%** | 85.36% | 82.35% | 86.47% |
| Repeated Median (Siegel, 1982) | 88.60% | 87.76% | 86.97% | 85.77% | 81.82% | 86.19% |
| FoolsGold (Fung et al., 2018) | 9.70% | 9.57% | 10.72% | 11.42% | 9.98% | 10.28% |
| Ours | **89.17%** | **88.60%** | 86.66% | **86.09%** | **85.81%** | **87.27%** |

Table 1: Results of label-flipping attacks on CIFAR-10 dataset with different numbers of attackers. The total number of participants is 10.

| # of attackers | 0 | 1 | 2 | 3 | 4 | Average |
|---|---|---|---|---|---|---|
| FedAvg (McMahan et al., 2017) | **91.81%** | 86.91% | 24.97% | 12.52% | 9.78% | 45.20% |
| Median (Yin et al., 2018) | 91.73% | **91.87%** | **91.79%** | 91.43% | 91.17% | 91.60% |
| Trimmed Mean (Yin et al., 2018) | **91.81%** | 91.82% | 91.82% | 91.49% | 91.26% | 91.64% |
| Repeated Median (Siegel, 1982) | 91.55% | 88.41% | 23.22% | 11.70% | 9.62% | 44.90% |
| FoolsGold (Fung et al., 2018) | 50.79% | 49.45% | 47.44% | 49.71% | 49.95% | 49.47% |
| Ours | 91.71% | 91.79% | 91.76% | **91.67%** | **91.38%** | **91.66%** |

Table 2: Results of label-flipping attacks on Amazon Reviews dataset with different numbers of attackers. The total number of participants is 10.

### 4.3 RESULTS ON BACKDOOR ATTACKS

For pixel-pattern backdoor attacks (Gu et al., 2019) in federated learning(Bagdasaryan et al., 2018), attackers manipulate their local models so that the learned global model predicts some backdoor target label for any image embedded with certain patterns. An example is shown in Figure 6 in the Appendix. The global model can behave normally for clean data. We choose "bird" in CIFAR-10 and "2" in MNIST as the backdoor target labels. Similarly, for the preprocessed LOAN dataset that contains 92 features, we manipulate 6 features by assigning certain large values to them and change the labels of manipulated data to "Does not meet the credit policy. Status:Fully Paid." The training data is mixed with manipulated data and clean data to fit both the backdoor task and the main task. We compare the performance of aggregation algorithms under two backdoor attack scenarios, which

| Dataset | MNIST | | | | CIFAR-10 | | | |
|---|---|---|---|---|---|---|---|---|
| | Naive approach | | Model replacement | | Naive approach | | Model replacement | |
| | Acc. | A.S.R | Acc. | A.S.R | Acc. | A.S.R | Acc. | A.S.R |
| FedAvg | **99.08%** | 99.71% | 98.75% | 17.85% | 87.44% | 99.91% | 69.72% | 38.59% |
| Median | 98.91% | **10.34%** | 98.87% | 10.35% | 88.58% | 73.06% | 87.22% | 10.01% |
| Trimmed Mean | 98.97% | **10.34%** | 98.81% | 10.34% | 88.38% | 84.56% | 87.30% | 9.85% |
| Repeated Median | 98.96% | 10.36% | 98.82% | 10.32% | 88.22% | 75.25% | **87.57%** | 9.79% |
| FoolsGold | 96.20% | 12.51% | 97.96% | **10.27%** | 10.00% | 0.00% | 10.00% | 0.00% |
| Our | 98.97% | 10.35% | **98.88%** | 10.31% | **88.89%** | **9.65%** | 87.43% | **9.56%** |

Table 3: Results of backdoor attacks on MNIST and CIFAR-10. There are 10 participants, 1 of whom is an attacker. We denote the accuracy as Acc. and the attack success rate as A.S.R.

| Standard deviation | 0% | 100% | 200% | 300% | Average |
|---|---|---|---|---|---|
| FedAvg (McMahan et al., 2017) | **93.17%** | 78.80% | 53.74% | 9.26% | 58.74% |
| Median (Yin et al., 2018) | 73.92% | 64.35% | 71.54% | 63.24% | 68.26% |
| Trimmed Mean (Yin et al., 2018) | 87.59% | 74.87% | 73.76% | 75.74% | 77.99% |
| Repeated Median (Siegel, 1982) | 61.86% | 75.53% | 67.76% | 73.63% | 69.69% |
| FoolsGold (Fung et al., 2018) | 86.72% | 90.31% | 86.96% | 89.94% | 88.48% |
| Ours | 90.06% | **90.91%** | **89.06%** | **92.28%** | **90.58%** |

Table 4: Results of Gaussian noise attacks on the MNIST dataset when $\epsilon$ is sampled from a Gaussian distribution with different standard deviations. There are 100 participants where 10 are attackers.

are called the naive approach and the model replacement in Bagdasaryan et al. (2018). For the naive approach, an attacker poisons its local model and submits the malicious update in every round. For the model replacement, an attacker only poisons in one round to embed some patterns into the global model, so the attacker needs to scale up its malicious update before submission. In our experiment, the malicious participant attacks in round 6 and scales up its update by 100. We run 200 rounds for MNIST and 100 rounds for CIFAR and LOAN.

Table 3 summarizes the results of backdoor attacks. Our method is the highest in terms of accuracy on MNIST under both backdoor attack scenarios. Moreover, on the more challenging CIFAR-10 dataset, our algorithm is the only one that can defend the naive approach backdoor attack. In Figure 4, we plot the attack success rates over time under different aggregation algorithms except for FoolsGold because it completely fails in both main and backdoor tasks. Intuitively, backdoor attacks can easily succeed under FedAvg, and other baselines slow down the process but still reach high attack success rate into over 70% within 100 rounds. Our algorithm effectively defends the attack and remains stable with 9.65% attack success rate when being attacked continuously for 100 rounds. On the LOAN dataset, our method achieves higher accuracy 94.50% (FedAvg 93.65%) and 0.00% attack success rate (FedAvg 99.71%) under the naive approach attack after 100 rounds. Similarly, our method has 95.06% accuracy (FedAvg 94.11%) and 0.00% attack success rate (FedAvg 98.96%) under model replacement attacks after 100 rounds. Other baselines also have 0.00% attack success rate in two attacking scenarios except FoolsGold, whose attack success rate is 99.96% under the naive approach attack.

It is also interesting to notice that all the baselines on the CIFAR-10 dataset for the naive backdoor attack approach while they perform fairly well on the MNIST dataset. This is because the model used for CIFAR-10 experiments, ResNet-18, is more complicated and has more parameters ($1000\times$) than the simple CNN used for the MNIST dataset. To explore the effect of complex models, we trained a ResNet-18 model on the MNIST dataset and performed the same backdoor attack on it. The results are shown in Figure 7 in the appendix. All the baselines fail to defend but our algorithm remains stable, which is similar with the result of CIFAR-10. Besides, most baseline methods (FedAvg, Median, Trimmed Mean, and Repeated Median) are coordinate-wise operations, while our algorithm accumulates a weight for each model rather than each parameter. We believe the model-wise reweighting scheme preserves the structure of the parameters and can perceive higher-level information.

### 4.4 RESULTS ON GAUSSIAN NOISE ATTACKS

In Gaussian noise attacks, we try to simulate real-world model corruption with 10 corrupted users at different scales of noise. Gaussian noise attacks are performed by multiplying each parameter by a scale $\epsilon$ sampled from a Gaussian distribution with a mean of $1$. In the experiment on MNIST, we have 100 participants where some may be attackers. Each participant has 300 images from a random digit and other 300 images from another random digit. Table 4 summarizes the results when $\epsilon$ is sampled from a Gaussian distribution with different standard deviations and there are 10 attackers. Our aggregation algorithm outperforms other approaches under Gaussian noise at multiple scales.

### 4.5 ALTERNATIVE SCHEMES

Though our algorithm is inspired by the reweighting scheme from IRLS (Rand R, 1997), some alternative linear estimators and weighting schemes can replace the original ones. In this section, we replaced the repeated median estimator in our algorithm with Theil-Sen estimator or median estimator and replaced the weighting scheme with a zero-mean Gaussian function where smaller residual means larger weight. We tuned the $\lambda$ (in the Gaussian weighting it is the variance $\sigma$), and the results are shown in table 7 in the appendix. The experiments are performed on the MNIST dataset with label-flipping attacks. Median and Theil-Sen estimators achieve similar results as the Repeated Median estimator, but they can be attacked in a few cases. For example, $\lambda = 5$ and $\delta = 0.01$ for Median estimator and $\lambda = 5$ and $\delta = 0.05$ for Theil-Sen estimator. Theil-Sen estimator, especially, only has a breakdown point of 29.3%, meaning that it will more easily be broken when the number of attackers increases. The alternative Gaussian weighting scheme is fairly robust against label-flipping attacks. This demonstrates that the robustness comes from our framework rather than intricate hyperparameter tuning.

## 5 CONCLUSION

Federated learning utilizes private data on multiple devices to train a global model, but the simple aggregation algorithm in federated learning is vulnerable to malicious attacks. To tackle this problem, we present a novel aggregation algorithm with residual reweighting. Our experiments on computer vision, natural language processing, and financial data show that our approach is robust to label-flipping, Gaussian noise, and backdoor attacks while prior aggregation methods are not. Our algorithm is easy to implement and readily incorporated into existing federated learning frameworks. We hope our proposed aggregation algorithm can make federated learning more practical and robust in the future. Our source code will be made public.

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

## A  APPENDIX

### A.1  PROOF

For simplicity, we consider the bound of the error rate for training a single-parameter model on $K$ devices, each storing $S$ IID samples of data. We define the parameter of the local model $i$ as $\hat{y}^{(i)}$. Suppose that training data points are sampled from some unknown distribution $\mathcal{D}$. Let $f(y; x)$ be a loss function of the parameter $y \in \mathcal{Y}$ associated with the data point $x$, where $\mathcal{Y}$ is the parameter space, and $F(y) := \mathbb{E}_{x \in \mathcal{D}}[f(y : x)]$. Our goal is to learn an optimal model defined by the parameter that minimizes the population loss:

$$\mu = \arg \min_{y \in \mathcal{Y}} F(y) \tag{15}$$

The parameter space $\mathcal{Y}$ is assumed to be convex and compact with diameter D, i.e., $||y - y'||_2 \leq D$, $\forall y, y' \in \mathcal{Y}$. Suppose that there are $K$ devices and $U$ of them are corrupted. We denote the set of adversarial devices as $\mathcal{B}$ and the corruption ratio $\alpha = \frac{U}{K}$. Based on our algorithm, the residuals can be simplified as $(\hat{y}^{(i)} - \tilde{y})$, where $\tilde{y}$ is the median of $\{\hat{y}^{(i)}\}$ for $i = 1, 2, ..., K$. Let $\widetilde{|r|}$ be the median of absolute residuals, i.e., $\widetilde{|r|} = \text{median}(|\hat{y}^{(i)} - \tilde{y}|)$. Then, the normalized residual can be expressed as

$$e^{(i)} = \frac{\hat{y}^{(i)} - \tilde{y}}{\gamma \widetilde{|r|}(1 + \frac{5}{K-1})}, \tag{16}$$

and the parameter confidence is defined as

$$z^{(i)} = \begin{cases} 1, & \text{if } \left|e^{(i)}\right| \leq \frac{\sqrt{2}\lambda}{\sqrt{K}} \\[2mm] \left|\frac{\sqrt{2}\lambda}{\sqrt{K}e^{(i)}}\right|, & \text{if } \frac{\sqrt{2}\lambda}{\sqrt{K}} < \left|e^{(i)}\right| \leq \left|\frac{\sqrt{2}\lambda}{\sqrt{K}\delta}\right| \\[2mm] 0, & \text{if } \left|e^{(i)}\right| > \left|\frac{\sqrt{2}\lambda}{\sqrt{K}\delta}\right| \end{cases} \tag{17}$$

Then we will prove that the error of the global model $\boldsymbol{M_{global}} := \frac{1}{\sum_{j=1}^{K} z^{(j)}} \sum_{i=1}^{K} z^{(i)} \hat{y}^{(i)}$ is bounded:

$$\left|\frac{1}{\sum_{j=1}^{K} z^{(j)}} \sum_{i=1}^{K} z^{(i)} \hat{y}^{(i)} - \mu\right| = \tilde{\mathcal{O}}(\frac{1}{\sqrt{S}} + \frac{1}{\sqrt{SK}} + \frac{1}{S} + \frac{1}{\sqrt{K}\delta}). \tag{18}$$

*Proof.* We adopt assumptions 1, 2, 3, and 6, Theorem 8 and Lemma 3 from Yin et al. (2018) here. We can separate Equation 18 into two sets with adversarial participant $\mathcal{B}$ and normal users $[K] \setminus \mathcal{B}$ where $[K] = \{1, 2, \ldots, K\}$.

$$\left|\frac{1}{\sum_{j=1}^{K} z^{(j)}} \sum_{i=1}^{K} z^{(i)} \hat{y}^{(i)} - \mu\right| \leq \frac{1}{\sum_{j=1}^{K} z^{(j)}} \sum_{i=1}^{K} z^{(i)} \left|\hat{y}^{(i)} - \mu\right|$$

$$\leq \frac{1}{\sum_{j=1}^{K} z^{(j)}} \left(\sum_{i \in [K] \setminus \mathcal{B}} z^{(i)} \left|\hat{y}^{(i)} - \mu\right| + \sum_{i \in \mathcal{B}} z^{(i)} \left|\hat{y}^{(i)} - \tilde{y} + \tilde{y} - \mu\right|\right)$$

$$\leq \max_{i \in [K] \setminus \mathcal{B}}\{\left|\hat{y}^{(i)} - \mu\right|\} + \frac{1}{\sum_{j=1}^{K} z^{(j)}} \sum_{i \in \mathcal{B}} z^{(i)} \left|\hat{y}^{(i)} - \tilde{y}\right| + |\tilde{y} - \mu| \,. \tag{19}$$

Now we state here the assumptions and theorems from Yin et al. (2018).

**Assumption 1** (Smoothness of f and F, Assumption 1 from Yin et al. (2018)). *For any $x \in \mathcal{D}$, the derivative of $f(y; x)$ is L-Lipschitz, and the function $f(y; x)$ is L-smooth.*

**Assumption 2** (Sub-exponential gradients, Assumption 6 from Yin et al. (2018)). *We assume that the derivative of $f(y; x)$ with respect to $y$ is v-sub-exponential.*

**Proposition 1** (Theorem 8 from Yin et al. (2018)). *Define the median of $y^{(i)}$:*

$$\tilde{y} = \text{median}\{y^{(i)} : i \in [K]\} \tag{20}$$

*Suppose that Assumption 1 holds, $F(y)$ is $\lambda_F$-strongly convex.Then, with probability at least $1 - \frac{4}{1+SKLD}$, we have*

$$|\tilde{y} - \mu| \leq \tilde{\mathcal{O}}\left(\frac{1}{SK} + \frac{1}{\sqrt{S}}\left(\alpha + \sqrt{\frac{\log(1 + SKLD)}{K(1-\alpha)}} + \frac{1}{\sqrt{S}}\right)\right) = \tilde{\mathcal{O}}(\frac{\alpha}{\sqrt{S}} + \frac{1}{\sqrt{SK}} + \frac{1}{S}) \tag{21}$$

**Proposition 2** (Lemma 3 from Yin et al. (2018)). *Suppose that the derivative of the loss function f is i.i.d. v-sub-exponential. Then with a high probability, we have*

$$\max_{i \in [K] \backslash \mathcal{B}}\{\left|\hat{y}^{(i)} - \mu\right|\} \leq \frac{2v}{\sqrt{S}}\sqrt{\log(1 + SKLD) + \log K} = \tilde{\mathcal{O}}(\frac{1}{\sqrt{S}}) \tag{22}$$

Yin et al. (2018) prove the difference between the median of models and the optimal model is bounded in Proposition 1, and the maximum difference between an honest model and the optimal model is bounded in Proposition 2 with a high probability, which are exactly the third and the first term respectively in Equation 19.

Now let us consider $\frac{1}{\sum_{j=1}^K z^{(j)}} \sum_{i \in \mathcal{B}} z^{(i)} \left|\hat{y}^{(i)} - \tilde{y}\right|$. For an attacker $a \in \mathcal{B}$, there are two cases:

- $|e^a| \leq \frac{\sqrt{2}\lambda}{\sqrt{K}}$. From Equation 16 and $|e^a| \leq \frac{\sqrt{2}\lambda}{\sqrt{K}}$, we have $\left|\hat{y}^{(a)} - \tilde{y}\right| \leq \frac{C}{\sqrt{K}}$, where $C = \gamma\sqrt{2}\lambda\widetilde{|r|}(1 + \frac{5}{K-1})$.

- $|e^a| > \frac{\sqrt{2}\lambda}{\sqrt{K}}$. If $|e^a| > \left|\frac{\sqrt{2}\lambda}{\sqrt{K}\delta}\right|$ then it can be eliminated from Equation 17. From Equation 16 and $|e^a| > \frac{\sqrt{2}\lambda}{\sqrt{K}}$, we have $\frac{C}{\sqrt{K}} < \left|\hat{y}^{(a)} - \tilde{y}\right| \leq \frac{C}{\sqrt{K}\delta}$.

Combine two cases, we know that $\left|\hat{y}^{(a)} - \tilde{y}\right| \leq \frac{C}{\sqrt{K}\delta}$, and we have the following inequality,

$$\frac{1}{\sum_{j=1}^K z^{(j)}} \sum_{a \in \mathcal{B}} z^{(a)}\left|\hat{y}^{(a)} - \tilde{y}\right| \leq \sup_{a \in \mathcal{B}}\left|\hat{y}^{(a)} - \tilde{y}\right| \leq \frac{C}{\sqrt{K}\delta}. \tag{23}$$

Therefore, we prove that

$$\left|\frac{1}{\sum_{j=1}^K z^{(j)}} \sum_{i=1}^K z^{(i)}\hat{y}^{(i)} - \mu\right| = \tilde{\mathcal{O}}(\frac{1}{\sqrt{S}} + \frac{1}{\sqrt{SK}} + \frac{1}{S} + \frac{1}{\sqrt{K}\delta}). \tag{24}$$

$\square$

## A.2 MODEL DETAILS

The models we use in the experiments are described here. The CNN model used for classifying MNIST images (LeCun et al., 1998) can be found in Table 5. CIFAR-10 is trained with Resnet-18(He et al., 2015). The learning rate for both MNIST classifier and CIFAR-10 classifier is 0.01 with a SGD optimizer. We train MNIST classifier for 200 synchronous rounds and CIFAR-10 classifier for 100 rounds, with batch size 128 and 64 respectively. The Amazon Reviews (Ruining and Julian, 2016) classifier FastText (Joulin et al., 2016) is a two-layer deep neural network where the first layer is an embedding layer of dimension 100 and the second layer is a fully connected layer of dimension 1. We use binary cross entropy loss to train the model with Adam optimizer (Kingma and Ba, 2014) and learning rate 0.05. The input dimension (vocabulary size) of the model is 25,000. For the preprocessed LOAN dataset, the input data consists of 91 features. We use a simple neural network with three fully-connected layers whose feature channels are 46, 23, and 9 respectively.

| Layer | # of Channels | Kernel size | Image size |
|---|---|---|---|
| Conv1 | 4 | 5x5 | 24x24 |
| Max-pool | 4 | | 12x12 |
| Conv1 | 8 | 5x5 | 8x8 |
| Max-pool | 8 | | 4x4 |
| fc1 | 16 | 16 | |
| fc2 | 10 | 10 | |

Table 5: The CNN architecture for MNIST classification.

### A.3 HYPERPARAMETERS

We have two hyperparameters in our method, $\lambda$ and $\delta$, where $\lambda$ controls the confidence interval, and $\delta$ controls the clipping threshold. We did a grid search to prove that the selection is robust and efficient. We conduct experiments on the MNIST dataset in i.i.d. and non-i.i.d. settings with various $\lambda$ and $\delta$. The results can be found in table 6. In the i.i.d. setting of the MNIST dataset, the results are almost the same for our selections of $\lambda$ and $\delta$. It is mainly because the parameters for honest models are very close, thus outliers are more easily to be excluded. Still, since the honest models are close and outliers can be easily detected, a larger $\delta$ yields a lower error bound as also shown in our proof. In the non-i.i.d. Experiments, however, the models are more diverse, so it is important to find a balance between excluding adversarial models while embracing models trained on different data distributions. Luckily, the hyperparameter is robust in a large range, as shown in table 6. Larger $\delta$ means it has a lower tolerance of differences and thus may eliminate some of the honest models when data is non-i.i.d. While increasing $\lambda$ brings insensitivity to the variance of $\delta$.

| $\lambda$ | $\delta$ | Number of attackers 0 | 5 | 9 | $\lambda$ | $\delta$ | Number of attackers 0 | 5 | 9 |
|---|---|---|---|---|---|---|---|---|---|
| 0.5 | 0.01 | 97.58% | 97.71% | 97.81% | 0.5 | 0.01 | 93.63% | 95.22% | 94.93% |
| | 0.05 | 97.70% | 97.00% | 97.39% | | 0.05 | 86.22% | 82.66% | 91.89% |
| | 0.1 | 97.68% | 97.38% | 97.76% | | 0.1 | 83.61% | 87.90% | 87.70% |
| | 0.2 | 97.74% | 97.15% | **98.05%** | | 0.2 | 79.87% | 87.64% | 88.86% |
| 1 | 0.01 | 97.72% | 97.68% | 97.44% | 1 | 0.01 | 94.41% | 94.92% | 94.54% |
| | 0.05 | 97.75% | 97.65% | 97.94% | | 0.05 | 93.36% | 93.99% | 91.15% |
| | 0.1 | 97.69% | 97.60% | 97.26% | | 0.1 | 86.93% | 92.20% | 89.39% |
| | 0.2 | 97.31% | 97.45% | 97.26% | | 0.2 | 84.77% | 84.91% | 91.40% |
| 2 | 0.01 | 97.59% | 97.68% | 97.75% | 2 | 0.01 | **94.95%** | 94.21% | 94.86% |
| | 0.05 | 97.64% | 97.85% | 97.51% | | 0.05 | 91.45% | 93.39% | 93.14% |
| | 0.1 | **97.94%** | **97.97%** | 97.76% | | 0.1 | 93.08% | 92.84% | 91.84% |
| | 0.2 | 97.39% | 97.26% | 97.98% | | 0.2 | 86.09% | 90.83% | 91.43% |
| 3 | 0.01 | 97.70% | 97.70% | 97.51% | 3 | 0.01 | 93.83% | 91.67% | 94.89% |
| | 0.05 | 97.50% | 97.90% | 97.56% | | 0.05 | 93.76% | **95.37%** | **95.68%** |
| | 0.1 | 97.39% | 97.49% | 97.65% | | 0.1 | 94.74% | 91.93% | 94.13% |
| | 0.2 | 97.44% | 97.79% | 97.57% | | 0.2 | 89.11% | 92.03% | 93.25% |
| 5 | 0.01 | 97.63% | 97.82% | 97.60% | 5 | 0.01 | 92.62% | 94.90% | 93.77% |
| | 0.05 | 97.83% | 97.61% | 97.86% | | 0.05 | 94.53% | 93.91% | 95.28% |
| | 0.1 | 97.69% | 97.68% | 97.65% | | 0.1 | 94.23% | 94.57% | 94.47% |
| | 0.2 | 97.26% | 97.55% | 97.39% | | 0.2 | 92.60% | 93.05% | 94.23% |

Table 6: Results of tuning hyperparameters $\lambda$ and $\delta$. All experiments are performed on MNIST dataset with label-flipping attacks. The left table is results with i.i.d. setting and the right table is with non-i.i.d. setting.

| $\lambda$ | | Median Estimator | | Theil-Sen Estimator | | Gaussian Weighting | |
|---|---|---|---|---|---|---|---|
| | | Number of attackers | | Number of Attackers | | Number of Attackers | |
| (or $\sigma$ in Gaussian) | delta | 0 | 9 | 0 | 9 | 0 | 9 |
| 1 | 0.01 | 94.46% | **95.19%** | 93.76% | 92.87% | 84.32% | 92.22% |
| 1 | 0.05 | 93.37% | 93.79% | 94.43% | 92.70% | 87.33% | 90.65% |
| 1 | 0.1 | 83.77% | 90.93% | 92.77% | 94.31% | 88.31% | 89.23% |
| 1 | 0.2 | 70.84% | 80.79% | 93.28% | 93.63% | 83.22% | 90.28% |
| 2 | 0.01 | 93.34% | 94.36% | 94.38% | 49.28% | 91.07% | 92.70% |
| 2 | 0.05 | 93.41% | 94.86% | **95.62%** | 91.65% | 90.85% | 93.00% |
| 2 | 0.1 | 94.02% | 93.48% | 92.29% | 93.07% | 88.61% | 93.15% |
| 2 | 0.2 | 88.84% | 92.68% | 92.21% | 91.70% | 90.54% | 90.80% |
| 3 | 0.01 | 94.67% | 94.68% | 94.45% | 75.83% | 92.46% | 93.18% |
| 3 | 0.05 | 93.67% | 94.52% | 94.86% | **94.72%** | 93.30% | **94.25%** |
| 3 | 0.1 | 93.11% | 91.30% | 92.32% | 94.70% | 92.09% | 93.65% |
| 3 | 0.2 | 93.67% | 93.76% | 94.00% | 93.20% | 90.88% | 93.26% |
| 5 | 0.01 | 93.68% | 84.26% | 94.69% | 93.27% | **94.10%** | 93.58% |
| 5 | 0.05 | 94.23% | 94.72% | 93.67% | 79.91% | 92.78% | 93.69% |
| 5 | 0.1 | **94.88%** | 94.69% | 94.60% | 92.85% | 92.81% | 93.83% |
| 5 | 0.2 | 92.90% | 93.87% | 93.51% | 91.41% | 91.72% | 92.93% |

Table 7: Results of replacing linear estimator and weighting schemes with other methods. All experiments are performed on MNIST dataset with label-flipping attacks.

## A.4 ALTERNATIVE SCHEMES

Though our algorithm is inspired by the reweighting scheme from IRLS (Rand R, 1997), some alternative linear estimators and weighting schemes can replace the original ones. In this section, we replaced the repeated median estimator in our algorithm with Theil-Sen estimator or median estimator and replaced the weighting scheme with a zero-mean Gaussian function where smaller residual means larger weight. We tuned the $\lambda$ (in the Gaussian weighting it is the variance $\sigma$), and the results are shown in table 7. The experiments are performed on the MNIST dataset with label-flipping attacks. Median and Theil-Sen estimators achieve similar results as the Repeated Median estimator, but they can be attacked in a few cases. For example, $\lambda = 5$ and $\delta = 0.01$ for Median estimator and $\lambda = 5$ and $\delta = 0.05$ for Theil-Sen estimator. Theil-Sen estimator, especially, only has a breakdown point of 29.3%, meaning that it will more easily be broken when the number of attackers increases. The alternative Gaussian weighting scheme is fairly robust against label-flipping attacks. This demonstrates that the robustness comes from our framework rather than intricate hyperparameter tuning.

## A.5 UNDERREPRESENTED DATA

We admit that underrepresented data samples might be rejected by our scheme. To investigate how it may affect the final result, we did a few experiments on the non-i.i.d. MNIST dataset. The experiment setting is the same as the label-flipping attack, where there are 100 users, and each of them holds images of 2 random digits (each digit has 20 holders). Now we remove the holders of digit 0 one by one until there are no user holding digit 0. The experiment result is plotted in figure 5. Our method performs significantly worse than FedAvg after removing 15 holders (75% images of digit 0). However, we observe similar effects on other algorithms, and we perform no worse than other algorithms.

## A.6 BACKDOOR EXPERIMENTAL SETUP

Experiment setup of backdoor attacks is provided here. There are 10 total participants each round where one of them is the attacker. In each round of federated learning, benign participants train their local models for 5 epochs and attackers train for 10 epochs. Benign participants always use a learning rate of 0.01. In MNIST and CIFAR, the attacker uses a learning rate of 0.1, and in LOAN, the attacker uses 0.002. The batch size is 64 for all participants. The attacker poisons 20 samples per batch in the training process. An example of backdoor patterns we use for image datasets is shown in Figure 6.

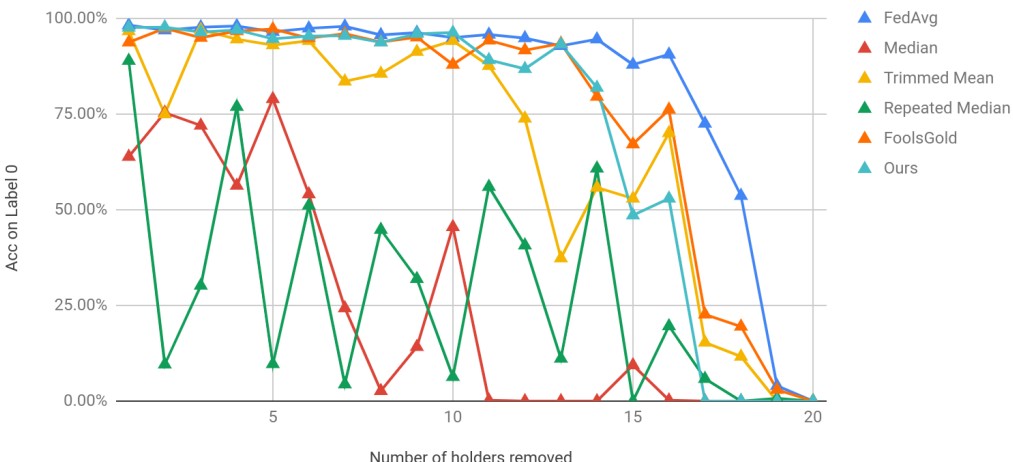

Figure 5: Results of removing users with digit 0.

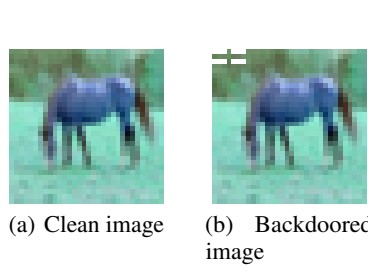

(a) Clean image    (b) Backdoored image

Figure 6: A backdoored image. There is a white color pattern in the left upper corner.

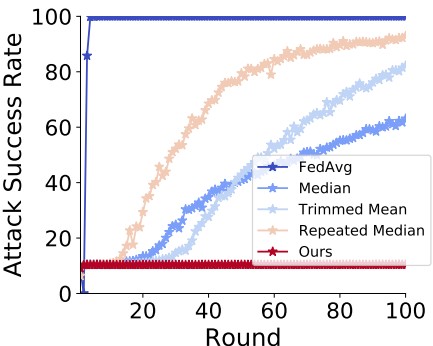

Figure 7: Results of backdoor attack success rate on the MNIST dataset with ResNet-18. Our algorithm is still stable while others fail for the complicated model.

For the LOAN dataset, we assign certain values that are slightly larger than their maximum value to the six features (num_tl_120dpd_2m, num_tl_90g_dpd_24m, pub_rec_bankruptcies, pub_rec, acc_now_delinq, tax_liens).

### A.6.1 DISCUSSION ON BACKDOOR ATTACK

While our algorithm can successfully defend pixel-pattern backdoor attacks (Gu et al., 2019), some interesting phenomena emerge in our experiments, as shown in table 3 and figure 4.

The first issue is the failure of all the baselines on the CIFAR-10 dataset for the naive backdoor attack approach while they perform fairly well on the MNIST dataset. This is because the model used for CIFAR-10 experiments, ResNet-18, is more complicated and has more parameters ($1000\times$) than the simple CNN used for the MNIST dataset. To explore the effect of complex models, we trained a ResNet-18 model on the MNIST dataset and performed the same backdoor attack on it. The results are shown in Figure 7 except for FoolsGold because it completely fails in both main task (about 10.00% accuracy) and backdoor task. All the baselines fail to defend but our algorithm remains stable, which is similar with the result of CIFAR-10. Besides, most baseline methods (FedAvg, Median, Trimmed Mean, and Repeated Median) are coordinate-wise operations, while our algorithm accumulates a weight for each model rather than each parameter. We believe the model-wise reweighting scheme preserves the structure of the parameters and can perceive higher-level information.

The second phenomenon is that the attack success rate starts increasing for the baseline methods only after around 25 iterations, as shown in Figure 4. The global model converges in the main task after 25

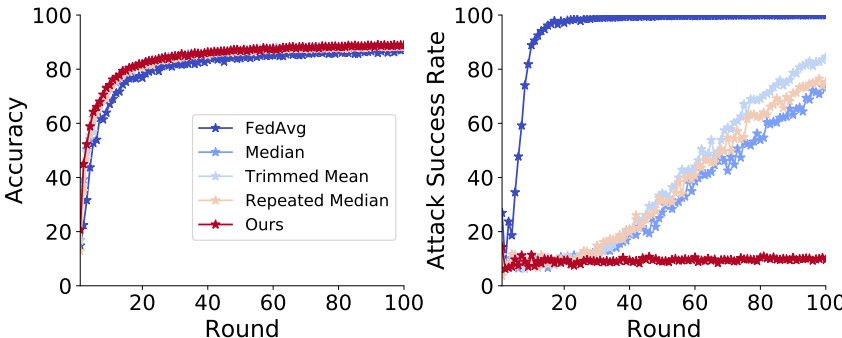

Figure 8: Accuracy of the global model and the attack success rate under backdoor attack. The attack success rate of baseline methods start to increase when the model converges.

iterations when the benign updates reflect more distinctive features of their local data (Bagdasaryan et al., 2018) and less shared common features. So the malicious updates are harder to be detected by the baselines after 25 iterations and the backdoor attack begins to success under baseline methods. We plotted the accuracy of the global model along with the attack success rate in Figure 8 to demonstrate the relation between model convergence and attack success rate.

### A.7 OTHER POTENTIAL ATTACKS

In this section, we discuss a potential attack to our algorithm. Because we accumulate confidence from parameters to the model confidence, an adversarial model may be able to accumulate enough confidence if the attacker changes part of the model to an honest model.

Assuming that attackers can observe the models from all users, they may bypass our algorithm. We evaluate this possibility by mixing adversarial models with honest models to different ratios. We perform this attack on the Amazon review dataset with 10 users, 4 of which are attackers. The attackers also train for 2 more epochs with learning rate 0.1 (the learning rate of regular users is 0.01).

The result is shown in table 8. The mix rate means the proportion of the honest model is changed to adversarial parameters. Therefore, larger mix rate indicates more adversarial parameters in the model (mix rate 1 means the model is entirely an adversary model). We use dropout to perform the blending, where the mix rate is the probability of dropping an honest parameter. This attack can be successfully defended by our algorithm.

| Mix rate | 0.5 | 0.3 | 0.1 | 0.03 | 0.01 | 0.001 | 0.0001 |
|----------|-----|-----|-----|------|------|-------|--------|
| Accuracy | 92.50% | 92.58% | 92.71% | 92.60% | 92.42% | 92.66% | 92.56% |

Table 8: The effect of blending adversarial models with honest models. Mix rate means to which portion the parameters of a model is changed to an adversarial model.

