# OpenReview forum: "Attack-Resistant Federated Learning with Residual-based Reweighting"
_ICLR.cc/2020/Conference — Reject_

### Official Review · AnonReviewer3 · 2019-10-21
**Official Blind Review #3**

**Rating:** 3

**Review:**

The paper proposes an algorithm for mitigating poisoning attacks in federated learning settings and compares it , on four different datasets, against state-of-the-art baselines.

Except for some minor issues (see the list below), the paper is well-written and -organized. The description of the proposed algorithm (in pseudo code and using the illustration in Figure 2) is very clear. Overall, the experiments are carefully described.

My main concern is that many choices in the design of the proposed algorithm lack context/discussion and thus appear rather ad-hoc. For instance,
- Why is the repeated median estimator used for estimating the linear regression? Could other robust estimators have been used?
- Similarly, could alternative weighting schemes be used in equations (3)-(5)?
I think it's important to provide more context and discuss possible alternatives. An important element is the exact threat model that the authors are considering. E.g., in the last paragraph on page 4, the authors mention specific attack strategies like altering only 10% of the model parameters. It appears that the design of the model weighting scheme aims at defending against these specific types of attacks. It will be good to either discuss or evaluate empirically how this scheme performs against other strategies.

The theoretical guarantee in Section 3.2 is a bit sketchy in my opinion. In what sense is $\mu$ the "expected value of the global model"? I.e. what is the expectation over? Consequently, I could not follow the statement in equation (14). Some explanation in plain text is needed here, too: in what sense does this equation provide a guarantee?

In the experiments, several aspects deserve further discussion: (1) the poor performance of FoolsGold almost across the entire board (except for the Gaussian noise attacks), which may indicate that this method was applied outside the threat model it was designed for; (2) the failure of all the baselines on CIFAR-10 for the naive attacking approach, while they perform fairly well on MNIST; (3) why does the attack success rate starts increasing in Figure 4 for the baseline methods only after ~25 iterations? (4) why do the baseline methods perform so poorly against label-flipping against on MNIST (Figure 3) while performing fairly well on CIFAR-10 and Amazon reviews (Table 1/2)? - I think that answering those questions may shed insights into the type of attacks that the different defences can / cannot withstand. I'd also like to challenge the authors to address whether they expect their defence to match or outperform the baselines on *any* attack strategy, or whether they can come up with scenarios where some of the baselines perform better? I would expect that the latter should be possible; it would not diminish the value of the proposed defence but shed more clarity on its possible limitations.

List of minor issues:
- in the abstract: "aggression" -> "aggregation"
- p.1: I would omit the statement in brackets "less than 100 lines".
- p.2: some of the related work discussion repeats content from the introduction
- p.3: "summaries" -> "summarizes"
- p.3: what does that mean: "has a high breakdown point of 50%"? Please explain/clarify.
- p.4: "is the k-the diagonal of matrix in Hn" -> "is the k-th diagonal element of the matrix Hn"
- p.4: my pdf reader couldn't render the binary operator on the right hand side of equation (8)
- p.5: "the details of the proof is presented" -> "are"
- p.8: "that of which" -> "whose"
- p.8: upper case "We" after comma
- p.8: first column "Acc" in Table 3: FedAvg has the highest accuracy. Generally, bold numbers in tables do not always mark the best-performing method. Sometimes, bold numbers are entirely missing. In cases where the difference is insignificant (which often appears to be the case) I would mark multiple numbers in bold, as appropriate.



**Experience Assessment:**

I have read many papers in this area.

**Review Assessment: Checking Correctness Of Derivations And Theory:**

I assessed the sensibility of the derivations and theory.

**Review Assessment: Checking Correctness Of Experiments:**

I assessed the sensibility of the experiments.

**Review Assessment: Thoroughness In Paper Reading:**

I read the paper at least twice and used my best judgement in assessing the paper.

---

### Official Review · AnonReviewer2 · 2019-10-23
**Official Blind Review #2**

**Rating:** 6

**Review:**

The paper proposes an aggregation algorithm, based on repeated median regression and residual-based weighting to defend federated learning from adversarial attacks. Experiments are shown to demonstrate the method robustness against label-flipping, backdoor and Gaussian noise attacks.

The paper is interesting, the topic recent and the methodology quite new, but a number of comments arise:

1) the methodology seems to rely on a number of ad-hoc steps, hyper parameter-dependent, that might hinder reproducibility and generalization: i) residual normalization via IRLS; ii) confidence assignment; iii) extreme value correction. Could be interesting to show analysis how varying such hyper-parameters affects the results, or otherwise to add further explanation at the comments in Appendix A.3 as to why the model seems to be insensitive to \lambda, or why \delta is significantly affected by data distribution.

2) could the repeated median estimator still be affected by the “federated size” i.e. the number of models involved in the federated learning? Is there any bound on the number of participants, below which the estimator would perform poorly?

3) proof of eq (14) could be more readable if
	•	full passages were shown (for instance for the reviewer the first passage was not immediate and took adding and subtracting \sum_i z^(i)\mu/\sum_i z^(i), and further simplification to be addressed), and
	•	ii) reference to the exact point in which previous results are used were made explicit (i.e. where in (Yin et al 2018) the bounds are proven).

4) proof of eq (14), when the attacker a\in B is fixed, then |\hat{y}^(i) - \tilde{y}| should be replaced by  |\hat{y}^(a) - \tilde{y}|

5) A last question concerns the aspect of "fairness" of this learning strategy. By removing aberrant updates there is still a chance of excluding from the learning process nodes that are intrinsically different form the average ones. In this sense, it is not clear from the paper how the reweighing strategy can mitigate this aspect, as there is no certainty that underrepresented data samples would not be rejected with the proposed scheme. Still aspect could have been better investigated in controlled scenarios.


**Experience Assessment:**

I have read many papers in this area.

**Review Assessment: Checking Correctness Of Derivations And Theory:**

I carefully checked the derivations and theory.

**Review Assessment: Checking Correctness Of Experiments:**

I carefully checked the experiments.

**Review Assessment: Thoroughness In Paper Reading:**

I read the paper at least twice and used my best judgement in assessing the paper.

---

### Official Review · AnonReviewer1 · 2019-10-24
**Official Blind Review #1**

**Rating:** 3

**Review:**

This paper presents an approach to robust federated learning that uses robust regression to weigh all the model parameter coefficients in order to achieve the robustness.

Specifically, for each coefficient in the model, a repeated median estimator is used to compute a linear regression fit, and the residual of each individual model's coefficient is normalized and used to compute a confidence score for that coefficient. Coefficients which have too large a confidence have their confidence reset to zero, to avoid the influence of outliers. The local model's coefficients are now aggregated using a weighted average with weights given by these confidence scores.

They compare their algorithm experimentally to reasonable baselines of model aggregation algorithms: the FedAvg algorithm, 3 recent robust FL algorithm, and an approach based on a standard robust regression estimator, using experiments on 4 different datasets and 4 different neural net architectures. They test the robustness of these algorithms to label flipping (MNIST, CIFAR-10), backdoor attacks, and multiplicative gaussian noise corruption of the model coefficients.

Overall the paper presents an interesting and novel approach to robustness in FL, using a robust regression estimator to aggregate the model coefficients. The motivation of the algorithmic design is for the most part clear, but the rationale behind the particular choice of the parameter confidence score is unclear, and should be clarified. The theory in support of the method seems reasonablish, but key definitions and steps in the proof are not explained in detail, referring instead of an earlier paper. In particular, it is not clear how the smoothness of the loss function and subexponentiality of its derivatives enter into the analysis of the method, nor do these parameters enter into the final error bounds. Also, how is mu defined in the error bound: what does it mean that it is the expected global model --- does this mean this would be the model if all the participating workers were non-corrupted, honest, and had iid data? This theory is hard to parse: more effort should be spent in clarifying the assumptions and definitions and showing how the claimed result follows. The specific result referenced from earlier work should be stated unambiguously as a proposition so the reader sees how it applies where it is used.

The experimental results show that the algorithm performs slightly better than the considered baselines in most situations considered, but the important question of the impact of hyperparameter selection for the method (e.g. the clipping at which the weights of "outlier" parameters are set to zero) and the competing methods (e.g. the clipping in the trimmed mean estimator) is not addressed-- the authors indicate that the method is robust to some choices and fixes them in the appendix. This makes it difficult to tell whether the method performs better due to careful or lucky hyperparameter selection.

Although the method is interesting and novel, and seems principled, the theoretical claims are unclear, and the experimental evaluation is not sufficiently informative about the impact of hyperparameter selection to draw conclusions about the effectiveness of this method of model aggregation as opposed to the baselines considered. In particular because of the latter issue, I'm leaning towards reject, but would be willing to change my score if this were addressed.

**Experience Assessment:**

I do not know much about this area.

**Review Assessment: Checking Correctness Of Derivations And Theory:**

I assessed the sensibility of the derivations and theory.

**Review Assessment: Checking Correctness Of Experiments:**

I assessed the sensibility of the experiments.

**Review Assessment: Thoroughness In Paper Reading:**

I read the paper at least twice and used my best judgement in assessing the paper.

---

### Author Response · Authors · 2019-11-15
**Summary of Revisions**

We want to thank the reviewers for their suggestions and comments! We have posted a revised version of the paper with several improvements based on the suggestions from the reviewers.

1. We revised our proof in A.1 to include the whole passage as well as the exact reference to the previous work.
2. We added more discussion and analysis in the experiment part 4.2 and 4.3 to help readers understand the intuition and design choices.
3. The revision also includes additional experiments on
  (1) hyperparameter selections in appendix A.3.
  (2) alternative linear estimator and weighting schemes in appendix A.4.
  (3) effects of underrepresented data in appendix A.5.
  (4) another potential attack in appendix A.7.
4. We added in-depth analysis in appendix A.6.1 to explain a few phenomena raised in the backdoor attack.

---

### Decision · Program_Chairs · 2019-12-19

**Decision:**

Reject

**Comment:**

The paper proposes an aggregation algorithm for federated learning that is robust against label-flipping, backdoor, and Gaussian noise attacks. The reviewers agree that the paper presents an interesting and novel method, however the reviewers also agree that the theory was difficult to understand and that the success of the methodology may be highly dependent on design choices and difficult-to-tune hyperparameters.